# Cultural and Linguistic Adaptation of the Fall Risk Questionnaire—Portuguese Version

**DOI:** 10.3390/ijerph20021598

**Published:** 2023-01-16

**Authors:** Ana Júlia Monteiro, Bernardo Constantino, Mariana Carvalho, Helena Silva, Ricardo Pedro, Rodrigo Martins, Tiago Atalaia, Jullyanne Silva, Pedro Aleixo, Sandra Alves

**Affiliations:** 1Physiotherapy Department, Escola Superior de Saúde da Cruz Vermelha Portuguesa–Lisboa, 1300-125 Lisboa, Portugal; 2Lusófona University, 1749-024 Lisbon, Portugal; 3MovLab–Biomechanics, CIDEFES–CICANT, Lusófona University, 1749-024 Lisbon, Portugal

**Keywords:** falls, risk assessment, validation study, elderly

## Abstract

Falls are a major problem among older adults worldwide. Early detection of risk factors is important to decrease the burden of falls. The “Fall Risk Questionnaire” (FRQ) is a screening tool used to assess personal fall risk awareness in older populations, and it is also used as a behavior change tool. The aim of the present study was to undertake a cultural and linguistic adaptation of the FRQ to allow its use in the Portuguese population. To achieve this, we used the sequential method proposed by Beaton and colleagues. Statistical analysis was conducted by computing the intraclass correlation coefficient and Cronbach’s alpha score for intra-observer reliability. The panel revision demonstrated high concordance for all assessed items. The pretesting results indicated that, in general, the FRQ—Portuguese Version (FRQ-VP) was well accepted, and all items were adequate and easy to comprehend. The ICC and Cronbach’s alpha scores indicated high consistency between assessments (*p* < 0.01; alfa = 0.992). The FRQ-VP seemed to have good reliability and internal consistency. Because the definition of a fall experience may not be clear to the older population, a careful explanation of this item can lead to a better score computation.

## 1. Introduction

Falls are a major problem among older adults worldwide, causing injuries that decrease quality of life and life span [1,2,3,4,5]. Early detection of factors that indicate increased risk of falls is one way to decrease the burden of falls [2,5,6] and avoid quality of life decrement in older populations. The recent pandemic scenario exposed the older population to greater solitude and inactivity, leading to increased fall risk in this population, as lower physical fitness seems to be related to falling [6,7]. One of the proposed risk indicators, and one of the strongest predictors of falling, is the existence of a fall history, with changes in gait or balance as the second most significant predictor [5]. As a first class predictor, a history of falls is very easy to factor into a prevention strategy through population screening, especially if it is integrated into a simple screening tool that aims to highlight risk factors [1,5,6,8,9]. This is because, most of the time, older people are not aware of their own risk of falling [2]. Screening with simple tools that can be applied quickly by health professionals or others can help to identify situations where this lack of awareness exists and clarify fall risk factor comprehension, leading to clarity regarding the actual fall risk of an older person [2]. Such tools can make population screening more efficient. However, to correctly assess populations, tools need to be valid and adequately adapted to the language and culture of the focus population [10]. The “Fall Risk Questionnaire” (FRQ) is a screening tool used to assess personal fall risk awareness in older populations, and it also serves as a behavior change tool [11,12]. The Geriatric Research, Education and Clinical Center, VA Greater Los Angeles Healthcare System, and their affiliates have been working to determine the beliefs of the elderly related to falls. This work led to the construction of the original FRQ, which aims to detect early fall risk and serve as a tool for preventive behavior enhancement and education [12]. The FRQ was validated against a “gold standard”—the clinical assessment proposed by the American/British Geriatrics Society Guidelines to assess independent fall risk predictors—with high agreement in the results [11]. The FRQ was then included in the brochure “Stay Independent” of the Stopping Elderly Accidents, Deaths & Injuries Program (STEADI) of the Center for Disease Control and Prevention (CDC) as the first step of the STEADI algorithm for screening assessment and intervention in the over 65 years elder community-dwelling population (see more at https://www.cdc.gov/steadi/index.html, accessed on 30 May 2022). The application of the FRQ provides important information for the clinical decision-making process, highlighting cases that may require care due to a lack of awareness of their own fall risk [11]. As a tool that comprises both awareness and behavior education, the FRQ aggregates important screening information into an easy-to-use tool, contributing to population empowerment and health literacy. This tool has already been adapted in Chinese, Turkish, Italian, and Thai languages, with good results [13,14,15,16].

The aim of the present study was the cultural and linguistic adaptation of the FRQ to allow its use in the older Portuguese population for screening purposes.

## 2. Materials and Methods

The process of cultural and linguistic adaptation of the FRQ utilized the sequential method proposed by Beaton and colleagues [10] and was preformed prior to collecting authorization from the original version’s authors to proceed. The sequential method comprises a sequence of phases the adapted version needs to pass to achieve validation. While not the only method, this method is widely used in cross-cultural adaptation studies (e.g., refs. [17,18]), and is also used in translations of the same questionnaire in other languages (e.g., ref. [14]).

Phase 1: Translation

The original version of the FRQ was translated by two independent, native Portuguese language translators. One of the translators had knowledge in the health area and the other did not. For each translation, the translators completed a report with their concerns and justification for their chosen translated terms.

Phase 2: Synthesis

The two translated versions (translation 1 [T1] and translation 2 [T2]) and associated reports were analyzed, and a synthesized version of the translation (T12) was constructed.

Phase 3: Back translation

The synthesized version was then back-translated by two native English language translators (back translation 1 [BT1] and back translation 2 [BT2]). Again, one had knowledge regarding the health area and the other did not. For each back translation, a report of concerns and justifications for the chosen terms was produced.

Phase 4: Expert committee review

The translations (T1, T2, and T12) and back translations (BT1 and BT2) were reviewed by an expert panel for semantic, idiomatic, experiential, and conceptual equivalence. The expert panel was composed of four translators and four physiotherapists. To collect the review criteria assessment, a Likert scale ranging from 0 to 5 was used. The rating of 0 was described as “do not understand”, 1 was “I understood almost nothing”, 2 was “I somewhat understood”, 3 was “I almost understood everything”, 4 was “I understood everything but have some questions”, and 5 was “I understood everything without questions”. Each expert provided a report with questions or suggestions. All the reports were analyzed and a final version of the Portuguese Version of the FRQ (FRQ-VP) was conceived.

Phase 5: Pretesting

The pretesting was conducted with a sample of 20 community-dwelling subjects, aged 65 or higher, all females, with full autonomy, without cognitive impairment as assessed by the Mini-Mental State Examination (MMSE), and without a disease history that could affect their cognitive capacity to comprehend the questions and the Portuguese context adaptation of the FRQ-VP.

To apply the FRQ-VP, two observers were trained in the application of both the FRQ-VP and MMSE. The observers read only the questions presented in the questionnaire. When finished the questionnaire, the observers were asked to give a report of the difficulties they observed during the application. The pretesting was conducted in a calm environment, without noise and with adequate lightning, to ensure minimal interpretation error due to environmental influence.

Phase 6: Original authors’ validation

The back translations, the FRQ-VP, and the reports produced were validated by the authors of the original version of the FRQ to affirm the original version’s attributes were maintained.

### 2.1. Reliability and Internal Consistency Verification

After the cultural and linguistic adaptation of the FRQ to create the FRQ-VP, this version was tested for reliability and internal consistency. To achieve this, a convenience sample of 89 subjects that were at least 65 years old was selected. The subjects were community-dwelling, with full autonomy, without cognitive impairment as assessed with the Mini-Mental State Examination (MMSE), and without relevant clinical history for the setting. Among the 89 subjects, 66 were women and 23 were men. All subjects filled in the FRQ-VP at two time points, with 10–14 days between them, to ensure that they had no significant memory issues and had not acquired new competences or experienced new fall events.

Observers were trained to apply the MMSE and to assist subjects with answering the FRQ-VP. Observers could only read the questions of the FRQ-VP if the subject did not know how to read or had physical handicaps that influenced their ability to independently answer the questionnaire. They could assist the subject in summing the final score. To collect data, subjects were scheduled and assessed individually by the observers. The same period of the day (morning or afternoon) was maintained in both assessments. The assessment was performed in a small room without distraction to ensure the best conditions for the application of the FRQ-VP and MMSE.

To test intra-observer reliability and internal consistency, the intraclass correlation coefficient (ICC) and Cronbach’s alpha score were used at a 95% confidence level. Based on the sample size requirements for estimating ICC presented by Bonett [19], for an estimated ICC of 0.8 at a 95% confidence level, a sample size of a minimum of 37 subjects was required. Statistical analysis was performed using the Statistical Package for Social Sciences (IBM SPSS software version 23, IBM, Armonk, NY, USA).

### 2.2. Ethical Considerations

All subjects who participated in this study signed informed consent after the study was explained to them and all of their questions had been answered, including their right to refuse to participate and their right to suspend participation at any part of the process.

This study was approved by the Ethical Commission of the Escola Superior de Saúde da Cruz Vermelha Portuguesa—Lisboa (number 05/2021).

## 3. Results

### 3.1. Semantic Equivalence

The results of the semantic equivalence are reported in Table 1.

The final version of the FRQ-VP can be observed in Figure 1.

### 3.2. Expert Panel Revision

The panel revision demonstrated high concordance for all assessed items. The lower concordance items were “Por vezes sinto-me inseguro (a) quando caminho/Insegurança ou a necessidade de apoio enquanto caminha são sinais de diminuição de equilíbrio” and “Ao caminhar em casa, sinto-me seguro (a) agarrando-me a móveis/Esta situação é, também, sinal de diminuição do equilíbrio”.

In the translation of the expression “I have fallen in the past year”, we chose to refer to the number of months to avoid any misunderstanding of the temporal window we wished to assess.

The panel suggested that the term “outros profissionais de saúde que o acompanham” (other health professionals that interact with you) be added to the item “Some o número de pontos (…) Discuta este folheto com o seu medico”. The panel justified this suggestion since not only doctors can refer elders with increased fall risk; all health professions with competencies in this matter can do so. The expression was then changed to “Some os pontos das respostas “Sim”. Se atingiu 4 pontos ou mais, pode estar em risco de queda. Discuta este folheto com o seu médico ou profissional de saúde que o acompanha”.

### 3.3. Pretesting

The pretesting sample was composed of 20 subjects, all female, aged between 69 and 94 years old (mean 80.45 ± 7.49 years), who were independent community-dwellers without cognitive impairment. Forty percent (40%) of the sample were illiterate and sixty percent (60%) had between 2 and 4 years of education (mean 3.5 ± 0.69 years of education). Sixty percent (60%) had a decrease in their hearing abilities, and fifteen percent (15%) presented with visual decline.

The subjects took 3 to 6 min to fill in the FRQ-VP (mean 4.15 ± 1.09 min) and the mean score was 8.75 ± 3.52 points (of 14 total). In general, the FRQ-VP was well accepted, and all items were considered to be adequate and easily comprehended.

### 3.4. Reliability and Internal Consistency Verification

The sample used to evaluate the reliability and internal consistency of the FRQ-VP was composed of 89 subjects aged between 65 and 91 years old (mean 76.21 ± 7.94 years). Female gender was prevalent (66 subjects, mean age 76.05 ± 8.06 years), leaving a total of 23 male subjects (mean age 76.7 ± 7.71 years). The social status of the subjects was 57.3% married, 33.71% widowed, 5.62% single, and 3.37% divorced. In total, 35.96% lived alone. Regarding living area characteristics, 51.69% lived in cities.

The data related to education level indicated a mean of 4.38 ± 3.61 years of education, ranging between 0 and 16 years. The rate of illiteracy was 19.1% and only 21.35% of the subjects had more than 4 years of education. Regarding reading and writing abilities, 83.15% could read and 77.53% could write.

In terms of health status, 86.52% of the subjects had multiple heath issues, with the most frequent being cardiovascular issues, musculoskeletal issues, diabetes, and decline of visual and additive acuity.

The selected sample presented a diverse and wide range of common characteristics among the elderly, which contributed to a very good understanding of the application of fall screening tools to this population.

### 3.5. Descriptive Results of the Portuguese Version of the Fall Risk Questionnaire

In the first application of the questionnaire, the final score had a mean value of 5.81 ± 3.56 (range 0–14). In the second application, collected 10 to 14 days later, the value obtained was 5.43 ± 3.35 (range 0–14). Table 2 shows the “yes” answers to each question for both administrations. Questions 1 and 10 registered the highest discrepancies between administrations.

Concerning the ICC and Cronbach’s alpha scores between the means of Observations 1 and 2, a strong correlation was found, indicating that the consistency between the assessments was high (Table 3). However, despite overall good consistency, differences were noted in the number of yes answers to Questions 1 and 10.

The effect size between observations was computed and is displayed in Table 4.

As we can see from the analysis given in Table 4, the effect size had a value of 0.236, representing a small effect size [20].

### 3.6. Results of Reliability and Internal Consistency

The ICC and Cronbach’s alpha score for each question of the FQR–VP are given in Table 5. Items 2, 9, and the final score had excellent values. On the other hand, items 1, 3, 4, 5, 6, 7, 11, and 12 had scores between moderate and high. Only item 10 had a score less than 0.7.

## 4. Discussion

Our results suggested that the Portuguese Version of the FRQ was consistent and could be used as a tool for self-reported fall risk screening. The process of cultural adaptation was similar to that used for other adaptations made on the same scale [14]. The methods presented by Beaton and colleagues [10] are not the only guidelines that exist; other guidelines, like those described by Sperber [21], are also valid guidelines that aim to produce culturally adapted instruments. Our choice of the Beaton and colleagues’ method was based not only on its widespread use for this purpose, including in recent literature [17], but also on its use of a pre-test phase that allows identification of population characteristics that can contribute to a better final version. This pre-testing provides an opportunity to see how much explanation observers need to use the scale, which could suggest the need for a change in the words or sentences used. No need for changes was indicated based on the pre-testing conducted in our study.

The reliability assessment of the FRQ-VP was conducted using the normal criteria from other studies that also translated the FRQ [13,14,15,16]. To address reliability, the ICC and Cronbach’s alpha score were used. The same approach was carried out in the Chinese, Turkish, and Thai translations, whereas the Italian translation used the Pearson coefficient. Comparing our results to the other translations that used the same statistical approach, we found that the Cronbach’s alpha score reported by the Chinese translation to be slightly smaller to that reported in our study. The Chinese version presented a range of Cronbach’s alpha score values between 0.602 and 0.720, while the range of our results was between 0.642 and 0.930. This indicated that our translation had a similar degree of comprehension by the respondents as the Chinese version. The same could be observed for the Turkish version; the results were similar when comparing the ICC scores for each question. The Turkish version presented a range of ICC scores between 0.954 and 1 and the Thai version presented an ICC value of 0.91, which were similar values to those reported in our version.

One of the items that we reported with a lower Cronbach’s alpha score was Question 10 (alpha = 0.642), meaning that differences were greater for this question. The importance given to medication among this population could have had some impact on this result. Additionally, the fact that this was reported as a fall-associated item could have made the subjects more aware of this relationship. This could be a point to address in future studies. In terms of the yes–no differences between observations, another question that also presented differences was Question 1, which referred to the number of falls the subject had experienced. One explanation for the differences could be that the understanding of a fall event was different between the two observations. The interaction with the professional that applied the FRS-VP could have contributed to a better reflection of fall events [2] and was probably related to a behavioral change towards the fall risk experience [11,12]. This difference did not present a notable effect size, as described by Cohen’s d of 0.236. None of the other translations noted this outcome.

The sample size seemed to have been adequate. A total of 203 participants were used in the Chinese version. In the Turkish version, 100 participants were used. The Thai version was conducted with 480 participants, and 176 participants were used in the Italian version. Considering that the original scale validation process was conducted with 40 participants using the criteria described by Bonett [19], our sample seemed adequate for the study purposes.

## 5. Conclusions

The FRQ-VP had good reliability and internal consistency. It was a suitable and easy to use self-reporting tool for fall risk assessment in the Portuguese population. Because the perception of a fall experience may differ among older people, a careful explanation of the definition of this item could lead to a better score computation.

## Figures and Tables

**Figure 1 ijerph-20-01598-f001:**
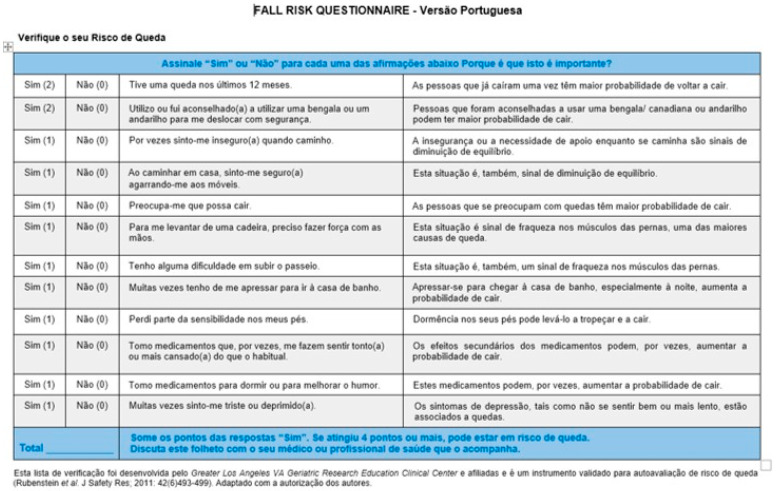
Final version of the Fall Risk Questionnaire—Versão Portuguesa (FRQ-VP) [11].

**Table 1 ijerph-20-01598-t001:** Semantic equivalence results. T—Translation; BT—Back Translation. T12 refers to the consensus version of the translation between T1 and T2.

Original Version	PT Translation	Back Translation	Final Version
Check Your Risk for Falling	T1-Verifique o seu Risco de Queda T2-Verifique o seu Risco de Queda T12-Verifique o seu Risco de Queda	BT1-Check your risk of falling BT2-Check Your Fall Risk	Verifique o seu Risco de Queda
Circle “Yes” or “No” for each statement below/Why it matters	T1-Circunde “Sim” ou “Não” para cada uma das afirmações abaixo/Por que razão estas afirmações são importantes T2-Selecione “Sim” ou ”Não” para cada frase abaixo./Qual a relevância? T12-Assinale “Sim” ou “Não” para cada uma das afirmações abaixo/Porque é que isto é importante?	BT1-Tick “Yes” or “No” for each of the statements below/Why is this important? BT2-Mark ‘Yes’ or ‘No’ for each statement below/Why does this matter?	Assinale “Sim” ou “Não” para cada uma das afirmações abaixo/Porque é que isto é importante?
I have fallen in the past year./People who have fallen once are likely to fall again.	T1-No ano passado, dei uma queda./As pessoas que já caíram uma vez têm probabilidade de voltar a cair. T2-Tive uma queda nos últimos 12 meses./Indivíduos que caíram pelo menos uma vez têm maior probabilidade de o voltar a fazer. T12-Tive uma queda nos últimos 12 meses./As pessoas que já caíram uma vez têm maior probabilidade de voltar a cair.	BT1-I have had a fall in the last 12 months./People who have already fallen once are likely to fall again. BT2-I have fallen during the past year./People that have fallen are more likely to fall again.	Tive uma queda nos últimos 12 meses./As pessoas que já caíram uma vez têm maior probabilidade de voltar a cair.
I use or have been advised to use a cane or walker to get around safely./People who have been advised to use a cane or walker may already be more likely to fall.	T1-Utilizo ou fui aconselhado(a) a utilizar uma bengala ou um andarilho para me deslocar com segurança./As pessoas que tenham sido aconselhadas a utilizar uma bengala ou um andarilho podem já ter estado em risco de queda. T2-Eu uso ou fui aconselhado a usar uma bengala ou andarilho para andar de forma segura./Indivíduos que foram aconselhados a usar uma bengala ou andarilho podem já ter maior probabilidade de cair. T12-Utilizo ou fui aconselhado(a) a utilizar uma bengala ou um andarilho para me deslocar com segurança./Pessoas que foram aconselhadas a usar uma bengala/canadiana ou andarilho podem ter maior probabilidade de cair.	BT1-I use or was advised to use a stick or walking frame to get around safely./People who have been advised to use a stick/crutch or walking frame may be more likely to fall. BT2-I use or have been advised to use a cane or walker to move around safely./People who have been advised to use a cane or walk have a higher probability of falling.	Utilizo ou fui aconselhado(a) a utilizar uma bengala ou um andarilho para me deslocar com segurança./Pessoas que foram aconselhadas a usar uma bengala/canadiana ou andarilho podem ter maior probabilidade de cair.
Sometimes I feel unsteady when I am walking./Unsteadiness or needing support while walking are signs of poor balance.	T1-Por vezes sinto-me inseguro(a) quando caminho./ A insegurança ou a necessidade de apoio enquanto se caminha são sinais de equilíbrio fraco. T2-As vezes sinto-me instável quando ando./Instabilidade ou a necessidade de suporte durante a marcha são sinais de equilíbrio reduzido. T12-Por vezes sinto-me inseguro(a) quando caminho./A insegurança ou a necessidade de apoio enquanto se caminha são sinais de diminuição de equilíbrio.	BT1-I sometimes feel insecure when I am walking./Insecurity or the need for support when walking are signs of a loss of balance. BT2-Sometimes I don’t feel steady when I walk./Feeling unsteady or needing support while you are walking are signs of poor balance.	Por vezes sinto-me inseguro(a) quando caminho./A insegurança ou a necessidade de apoio enquanto se caminha são sinais de diminuição de equilíbrio.
I steady myself by holding onto furniture when walking at home./This is also a sign of poor balance.	T1-Ao caminhar em casa, sinto-me seguro(a) agarrando-me aos móveis./Esta situação é, também, sinal de equilíbrio fraco. T2-Para me equilibrar, agarro-me à mobília em casa quando ando./Isto também é sinal de equilíbrio reduzido. T12-Ao caminhar em casa, sinto-me seguro(a) agarrando-me aos móveis./Esta situação é, também, sinal de diminuição de equilíbrio.	BT1-I feel safer holding onto the furniture when moving around the house./This situation is also a sign of a loss of balance. BT2-I feel safer holding onto furniture when I am walking at home./This situation is also a sign of poor balance.	Ao caminhar em casa, sinto-me seguro(a) agarrando-me aos móveis./Esta situação é, também, sinal de diminuição de equilíbrio.
I am worried about falling./People who are worried about falling are more likely to fall.	T1-Preocupo-me com quedas./As pessoas que se preocupam com quedas têm maior probabilidade de cair. T2-Preocupa-me que possa cair./Indivíduos que se preocupam com uma possível queda têm maior probabilidade de cair. T12-Preocupa-me que possa cair./As pessoas que se preocupam com quedas têm maior probabilidade de cair.	BT1-I worry I might fall./People who worry about falling are more likely to fall. BT2-I am worried about falling./People who worry about falling are more likely to fall.	Preocupa-me que possa cair./As pessoas que se preocupam com quedas têm maior probabilidade de cair.
I need to push with my hands to stand up from a chair./This is a sign of weak leg muscles, a major reason for falling.	T1-Para me levantar de uma cadeira, preciso de fazer força com as mãos./Esta situação é sinal de fraqueza nos músculos das pernas, causa potenciadora de quedas. T2-Preciso de usar as mãos para me levantar da cadeira./Isto é um sinal de fraqueza nos músculos das pernas, uma das maiores causas de quedas. T12-Para me levantar de uma cadeira, preciso de fazer força com as mãos./Esta situação é sinal de fraqueza nos músculos das pernas, uma das maiores causas de queda.	BT1-I have to push with my hands to get up from the chair./This situation is a sign of weakness in the leg muscles, one of the main causes of falls. BT2-To stand up from a chair I need to push with my hands./This is a sign of weak leg muscles, one major reason for falling.	Para me levantar de uma cadeira, preciso de fazer força com as mãos./Esta situação é sinal de fraqueza nos músculos das pernas, uma das maiores causas de queda.
I have some trouble stepping up onto a curb./This is also a sign of weak leg muscles.	T1-Tenho alguma dificuldade em subir um lancil de passeio./Esta situação é, também, sinal de fraqueza nos músculos das pernas. T2-Tenho alguma dificuldade em subir o passeio./Isto é, também, um sinal de fraqueza nos músculos das pernas. T12-Tenho alguma dificuldade em subir o passeio./Esta situação é, também, um sinal de fraqueza nos músculos das pernas.	BT1-I have some difficulty stepping up onto the sidewalk./This situation is also a sign of weakness in the leg muscles. BT2-I have some difficulty stepping onto the pavement./This is also a sign of weak leg muscles.	Tenho alguma dificuldade em subir o passeio./Esta situação é, também, um sinal de fraqueza nos músculos das pernas.
I often have to rush to the toilet./Rushing to the bathroom, especially at night, increases your chance of falling.	T1-Muitas vezes tenho de me deslocar apressadamente à casa de banho./Deslocar-se apressadamente à casa de banho, sobretudo à noite, aumenta a probabilidade de queda. T2-Muitas vezes tenho de me apressar para chegar a tempo à casa de banho./Apressar-se para chegar à casa de banho, especialmente à noite, aumenta a probabilidade de cair. T12-Muitas vezes tenho de me apressar para ir à casa de banho./Apressar-se para chegar à casa de banho, especialmente à noite, aumenta a probabilidade de cair.	BT1-I often have to rush to the toilet./Rushing to the toilet, especially at night, increases the probability of falling. BT2-I often have to rush to the toilet./Rushing to get to the toilet, especially at night, increases your chance of falling.	Muitas vezes tenho de me apressar para ir à casa de banho./Apressar-se para chegar à casa de banho, especialmente à noite, aumenta a probabilidade de cair.
I have lost some feeling in my feet./Numbness in your feet can cause stumbles and lead to falls.	T1-Perdi alguma sensibilidade nos pés./O entorpecimento dos pés pode causar tropeções e provocar quedas. T2-Perdi parte da sensibilidade nos meus pés./Dormência nos seus pés pode levá-lo a tropeçar e a cair. T12-Perdi parte da sensibilidade nos meus pés./Dormência nos seus pés pode levá-lo a tropeçar e a cair.	BT1-I have lost some feeling in my feet./Numbness in your feet may make you trip and fall. BT2-I have lost some feeling in my feet./Feet numbness can lead you to trip and fall.	Perdi parte da sensibilidade nos meus pés./Dormência nos seus pés pode levá-lo a tropeçar e a cair.
I take medicine that sometimes makes me feel light-headed or more tired than usual./Side effects from medicines can sometimes increase your chance of falling.	T1-Tomo medicamentos que, por vezes, me fazem sentir tonto(a) ou mais cansado(a) do que o habitual./Os efeitos colaterais dos medicamentos podem, por vezes, aumentar a probabilidade de queda. T2-Tomo medicamentos que por vezes me deixam com tonturas ou mais cansado do que o normal./Os efeitos colaterais dos medicamentos podem, por vezes, aumentar a probabilidade de queda. T12-Tomo medicamentos que, por vezes, me fazem sentir tonto(a) ou mais cansado(a) do que o habitual./Os efeitos secundários dos medicamentos podem, por vezes, aumentar a probabilidade de cair.	BT1-I take medication that sometimes makes me feel dizzy or more tired than usual./The side effects of medication can sometimes increase the probability of falling. BT2-I take some medicine that sometimes makes me feel dizzy or more tired than usual./Side effects from medication can sometimes increase your chance of falling.	Tomo medicamentos que, por vezes, me fazem sentir tonto(a) ou mais cansado(a) do que o habitual./Os efeitos secundários dos medicamentos podem, por vezes, aumentar a probabilidade de cair.
I take medicine to help me sleep or improve my mood./These medicines can sometimes increase your chance of falling.	T1-Tomo medicamentos para dormir ou para melhorar o humor./Estes medicamentos podem, por vezes, aumentar a probabilidade de queda. T2-Tomo medicação para me ajudar a dormir ou a melhorar a minha disposição./Estes medicamentos podem por vezes aumentar a probabilidade de cair. T12-Tomo medicamentos para dormir ou para melhorar o humor./Estes medicamentos podem, por vezes, aumentar a probabilidade de cair.	BT1-I take medication to sleep or to lift my mood./This medication can sometimes increase the probability of falling. BT2-I take some medicine to sleep or improve my mood./These medicines can sometimes increase your chance of falling.	Tomo medicamentos para dormir ou para melhorar o humor./Estes medicamentos podem, por vezes, aumentar a probabilidade de cair.
I often feel sad or depressed./Symptoms of depression, such as not feeling well or feeling slowed down, are linked to falls.	T1-Sinto-me triste ou deprimido(a) com frequência./Os sintomas de depressão, tais como não se sentir bem ou ter lentidão de movimentos, estão associados a quedas. T2-Muitas vezes sinto-me triste ou deprimido./Sintomas de depressão, tais como não se sentir bem ou mais lento, estão ligados a quedas. T12-Muitas vezes sinto-me triste ou deprimido(a)./Sintomas de depressão, tais como não se sentir bem ou mais lento, estão associados a quedas.	BT1-I often feel sad or depressed./Symptoms of depression, such as feeling down or slow, are associated to falls. BT2-I often feel sad or depressed./Depression symptoms, such as not feeling well or feeling that you have slowed down, are associated with falls.	Muitas vezes sinto-me triste ou deprimido(a)./Sintomas de depressão, tais como não se sentir bem ou mais lento, estão associados a quedas.
Add up the number of points for each “yes” answer. If you scored 4 points or more, you may be at risk for falling. Discuss this brochure with your doctor.	T1-Some os pontos das respostas “Sim”. Se atingiu 4 pontos ou mais, pode estar em risco de queda. Analise este folheto com o seu médico. T2-Some o número de pontos para cada resposta “sim”. Se o total for de 4 ou mais pontos, poderá estar em risco de queda. Discuta este panfleto com o seu medico. T12-Some o número de pontos para cada resposta “sim”. Se o total for de 4 ou mais pontos, poderá estar em risco de queda. Discuta este folheto com o seu medico.	BT1-Add up the “Yes” answers. If you reach 4 points or more, you may be at risk of falling. Discuss this leaflet with your doctor. BT2-Add up your points for every ‘Yes’ answer. If you scored 4 points or more, you may be at risk of falling. Discuss this leaflet with your doctor.	Some os pontos das respostas “Sim”. Se atingiu 4 pontos ou mais, pode estar em risco de queda. Discuta este folheto com o seu médico ou profissional de saúde que o acompanha.

**Table 2 ijerph-20-01598-t002:** Answers to each question. O 1—Observation 1; O 2—Observation 2.

		Q 1	Q 2	Q 3	Q 4	Q 5	Q 6	Q 7	Q 8	Q 9	Q 10	Q 11	Q 12
O 1	Mean	0.94 ± 1	0.63 ± 0.93	0.52 ± 0.5	0.28 ± 0.45	0.75 ± 0.43	0.43 ± 0.5	0.39 ± 0.49	0.44 ± 0.5	0.34 ± 0.48	0.18 ± 0.39	0.40 ± 0.49	0.51 ± 0.5
	Answers	42 (yes) 47 (no)	28 (yes) 61 (no)	46 (yes) 43 (no)	25 (yes) 64 (no)	67 (yes) 22 (no)	38 (yes) 51 (no)	35 (yes) 54 (no)	39 (yes) 50 (no)	30 (yes) 59 (no)	16 (yes) 73 (no)	36 (yes) 53 (no)	45 (yes) 44 (no)
O 2	Mean	0.65 ± 0.94	0.52 ± 0.88	0.53 ± 0.5	0.30 ± 0.46	0.80 ± 0.4	0.46 ± 0.5	0.37 ± 0.49	0.45 ± 0.5	0.30 ± 0.46	0.11 ± 0.32	0.43 ± 0.5	0.51 ± 0.5
	Answers	29 (yes) 60 (no)	23 (yes) 66 (no)	47 (yes) 42 (no)	27 (yes) 62 (no)	71 (yes) 18 (no)	41 (yes) 48 (no)	33 (yes) 56 (no)	40 (yes) 49 (no)	27 (yes) 62 (no)	10 (yes) 79 (no)	38 (yes) 51 (no)	45 (yes) 44 (no)

**Table 3 ijerph-20-01598-t003:** ICC and Cronbach’s alpha score between Observation 1 (O 1) and Observation 2 (O 2).

	Intraclass Correlation	Sig.	Cronback’s Alfa
Single measures	0.983	0.000	0.992
Mean Measures	0.992	0.000	

**Table 4 ijerph-20-01598-t004:** Effect size analysis by Cohen’s d between Observation 1 (O 1) and Observation 2 (O 2).

		Standardizer	Point Estimate.
Pair O 1–O 2	Cohen’s d	1.620022	0.236
	Hedges’ correction	1.63420	0.234

**Table 5 ijerph-20-01598-t005:** ICC and Cronbach’s alpha score between O 1 and O 2.

Question	ICC		Cronbach’s Alpha Score
	Mean Measure	Sig.	
FRQ_01	0.827	0.000	0.847
FRQ_02	0.927	0.000	0.930
FRQ_03	0.860	0.000	0.859
FRQ_04	0.807	0.000	0.806
FRQ_05	0.764	0.000	0.764
FRQ_06	0.828	0.000	0.828
FRQ_07	0.896	0.000	0.895
FRQ_08	0.858	0.000	0.857
FRQ_09	0.902	0.000	0.902
FRQ_10	0.637	0.000	0.642
FRQ_11	0.899	0.000	0.899
FRQ_12	0.815	0.000	0.813
FRQ_TOTAL	0.939	0.000	0.942

## Data Availability

Data collected by this study is not available due to privacy and ethical restrictions.

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
