# Peer review of "Cultural and Linguistic Adaptation of the Fall Risk Questionnaire—Portuguese Version"

_ijerph, 2023, doi:10.3390/ijerph20021598_

Round 1
Reviewer 1 Report
Good article! There some doubts about
sample size- is small number of females
the discussion part of the methods
conclusion needs to improve
Author Response
Reply to Reviewer 1.
Point 1: Good article!
Reply: Thanks for your support.
Point 2: There some doubts about sample size- is small number of females
Reply: Thanks for your comment. We understand that your question was regarding the sample size. We added a reference that justifies that for a 95% confidence interval, and expecting a ICC of 0,8, a sample size of 37 subjects is required. The question regarding the number of females, the population that we have access is manly composed by women. Because of the aim of the study, that factor do not seem to implicate any problem as we only assessed the consistency between the two assessments.
Point 3: the discussion part of the methods
Reply: Thanks for your comment. We understood that improvements need to be made on the discussion regarding the confrontation of our methods with other manuscripts that aim the same objectives. The modifications were made including other reviewers’ contributions. Hope it meet your expectations.
Point 4: conclusion needs to improve
Reply: Thanks for your comment. Because of the previous comment and the other reviewers’ comments, changes were made in the conclusions. Hope it meet your expectations.
Thanks for the comments and suggestions that made us improve our manuscript.
Best regards,
Reviewer 2 Report
With regard to manuscript: Cultural and Linguistic Adaptation of the Fall Risk Questionnaire – Portuguese Version, for consideration in Int. J. Environ. Res. Public Health. This is a very interesting manuscript addressing the falls in the life of older adults. The paper will contribute to knowledge and is worthy of publication, however, a clear discussion of the study is missing and has to be included. In addition, the results could be much better presented and explored. I have comments and questions that I have detailed below. With the utmost respect, allow me to give you a few suggestions.
· The novelty of the study could be more highlighted (in introduction). Cultural and linguistic adaptation of the FRQ, to allow it use in the Portuguese. No similar attempt has been made to other language? The authors would explain why their findings aggregates to the existing knowledge.
· Line 56 the word “is” appears strange.
· About correct validation (the sequence of phases: Translation, Synthesis… ), has this been previously reported or is there any scientific evidence? It was cited only the study of Beaton and colleagues. Is this study the best suited to the cultural and linguistic adaptation?
· Why was (only) female studied (especially in Phase 5 – Pretesting)? Falls are also a problem for male older. How do your findings translate to the male subjects? Social status has been considered, but health markers were not described (fat index, glycemia). It is necessary to consider that body composition (muscle mass) differs between males and females (at the age studied here). Further, I would like this point of view to be more in-depth in discussion. I understand the difficult for collecting information in humans, however I strongly believe that these limitations should be assumed.
· About the semantic equivalence results, my only observation is in the sentence “Tenho alguma dificuldade em subir o passeio” The term “passeio” could be misinterpreted in some countries that use Portuguese. This word could be replaced with words such as " guia, calçada ".
· Please give more details on original authors validation (Phase 6).
· The results should be more explored. As suggestion: To perform the absolute and percentage variation (∆) over time. It would be interesting report the effect size (Cohen's d).
· The meaning of words (T1, T2, T12, RT1, RT2) from table1 is not clear. These words should be explained in table caption.
· There are 12 questions, but why the range 0-14?
· To evaluate reliability and internal consistence of the FRQ-VP was composed by 89 subjects. It would be interesting report (in table) the NO answers to each question (to be clearer for the reader). This the sum of answers should be equal to 89. For example: Q1: 42 (Yes) 47 (No) for test and 29 (Yes) 60 (No) for retest.
· O1 and O2 is very confusing. Rewording is needed. Consider using test vs retest
· Didactics would improve with the inclusion a figure (some scheme drawn by the authors) explaining the 12 domains from FRQ-VP.
· Describe the experimental conditions in more detail. Give more details on FRQ-VP application, place of evaluations, time of day, instructions, observers, individual or collective evaluation? and others minor things.
· It is unclear how Cronback Alpha were calculated. Authors must include the mathematical formulation.
Author Response
Reply to Reviewer 2
Comment: With regard to manuscript: Cultural and Linguistic Adaptation of the Fall Risk Questionnaire – Portuguese Version, for consideration in Int. J. Environ. Res. Public Health. This is a very interesting manuscript addressing the falls in the life of older adults. The paper will contribute to knowledge and is worthy of publication, however, a clear discussion of the study is missing and has to be included. In addition, the results could be much better presented and explored. I have comments and questions that I have detailed below. With the utmost respect, allow me to give you a few suggestions.
Reply: Thanks for the compliment and for the suggestions. Mainly on this comment we retain the need to improve discussion and the presentation of the results. Some improvements were made using your suggestions that follows this initial comment. We’ll proceed with the reply for each point. Thanks again for your suggestions and comments.
Comment: The novelty of the study could be more highlighted (in introduction).
Reply: Thanks for your comment. We introduce more information on introduction that hope meets your expectations. We believe that improved that section. Thanks again for the suggestions.
Comment: Cultural and linguistic adaptation of the FRQ, to allow it use in the Portuguese. No similar attempt has been made to other language? The authors would explain why their findings aggregates to the existing knowledge.
Reply: Thanks for your comment. We included the translations that we were able to find, Chinese, Turkish, Thai and Italian. No other language seems to have a translation of it. The introduction and discussion were changed accordingly.
Comment: Line 56 the word “is” appears strange.
Reply: an extensive English revision was made, and all errors corrected. Thanks for your attention.
Comment: About correct validation (the sequence of phases: Translation, Synthesis… ), has this been previously reported or is there any scientific evidence? It was cited only the study of Beaton and colleagues. Is this study the best suited to the cultural and linguistic adaptation?
Reply: The Beaton and colleagues’ guidelines are one of the most used guidelines for the process of cultural adaptation. We included 2 more references that used the same methods and are recent. There are other methods, like Sperber A.D. DeVellis R.F. Boehlecke B. Cross-cultural translation methodology and validation. J Cross-Cult Psychol. 1994; 25: 501-524, but all centers in the main phases as Beaton and with similar methods. We included this remarks in the methods and discussion as believe that the justification is clearer. Thanks.
Comments: Why was (only) female studied (especially in Phase 5 – Pretesting)? Falls are also a problem for male older.
Reply: Thanks for your comment. The pretesting was made from a sample of 20 subjects, all females that were a sample that was more accessible. The fact that only women composed the sample do not constitute a problem as the aim of the study was to address the comprehension of the translated version and it consistency between observations, not with the aim to collect data clearly to understand fall risk. The results of this application regarding that issue will be part of other manuscript.
Comment: How do your findings translate to the male subjects?
Reply: We believe that the answer is somewhat already given in the reply of the previous comment.
Comment: Social status has been considered, but health markers were not described (fat index, glycemia). It is necessary to consider that body composition (muscle mass) differs between males and females (at the age studied here). Further, I would like this point of view to be more in-depth in discussion. I understand the difficult for collecting information in humans, however I strongly believe that these limitations should be assumed.
Reply: Thanks for your comment. This data was collected and will be part of other manuscript. Because the aim of the present study was to address the consistency of the reply made by the same subject in two different observation 10-14 days apart, this data wasn’t included. The social descriptive were included because they describe a pre-test sampling of very diverse elder characteristics that we consider to be of value to understand the application of the fall screening tool we wish to verify.
Comment: About the semantic equivalence results, my only observation is in the sentence “Tenho alguma dificuldade em subir o passeio” The term “passeio” could be misinterpreted in some countries that use Portuguese. This word could be replaced with words such as " guia, calçada ".
Reply: Thanks for your comment. This was the term that received the consensus from the expert panel. Thanks again for your suggestion.
Comment: Please give more details on original authors validation (Phase 6).
Reply: We improved the text on phase 6, providing a clearer information regarding this issue.
Comment: The results should be more explored. As suggestion: To perform the absolute and percentage variation (∆) over time. It would be interesting report the effect size (Cohen's d).
Reply: Thanks for the suggestion. We add more information and ad a discussion part. Hope it meet your expectations. We believe your suggestion improved the manuscript. Thanks.
Comment: The meaning of words (T1, T2, T12, RT1, RT2) from table1 is not clear. These words should be explained in table caption.
Reply: Thanks for your suggestion. It was performed accordingly.
Comment: There are 12 questions, but why the range 0-14?
Reply: Because two items of the FRQ pointed more than 1 point. To better understanding of this, we include a final version of the FRQ-VP in figure 1.
Comment: To evaluate reliability and internal consistence of the FRQ-VP was composed by 89 subjects. It would be interesting report (in table) the NO answers to each question (to be clearer for the reader). This the sum of answers should be equal to 89. For example: Q1: 42 (Yes) 47 (No) for test and 29 (Yes) 60 (No) for retest.
Reply: Thanks for the suggestion. We change accordingly to it.
Comment: O1 and O2 is very confusing. Rewording is needed. Consider using test vs retest
Reply: Thanks for the suggestion. We include the legend, and we believe the term is more clearly understood.
Comments: Didactics would improve with the inclusion a figure (some scheme drawn by the authors) explaining the 12 domains from FRQ-VP.
Reply: We included the final version of the FRQ – VP that we believe achieve your suggestion. Thanks for your comment.
Comments: Describe the experimental conditions in more detail. Give more details on FRQ-VP application, place of evaluations, time of day, instructions, observers, individual or collective evaluation? and others minor things.
Reply: we improved the description presented on page 3.
Comment: It is unclear how Cronback Alpha were calculated. Authors must include the mathematical formulation.
Reply: Thanks for the suggestion. Cronback’s Alpha is automatic computed by SPSS as we preform the ICC analysis. If with the changes made you still feel that the mathematical formulation of the Cronback’s alpha is needed we are open to include it, but the option to not include the mathematical formulation follows the same option in other studies that aims the same objective as ours. Hope you agree with our option. Thanks for your comment.
Reviewer 3 Report
Dear author,
I find this article very interesting, but it needs some improvements.
Sincerely
please find the attachment

Author Response
Reply to Reviewer 3
Comment: Review reports should contain the following:
A brief summary (one short paragraph) outlining the aim of the paper, its main contributions and strengths.
Cultural and Linguistic Adaptation of the Fall Risk Questionnaire - Portuguese Version is an original research paper describing the cultural and linguistic adaptation of the Falls Risk Questionnaire (FRQ) to screen and appropriately assess the population at risk.
Reply: Thanks for your compliment and comment.
General concept comments
Comment: Article: highlighting areas of weakness, the testability of the hypothesis, methodological inaccuracies, missing controls, etc.
This article is an original research paper entitled: Cultural and Linguistic Adaptation of the Fall Risk Questionnaire - Portuguese version with the main aim of making it useful for the Portuguese population. In the introduction, the author describes the importance of screening the population for falls in old age, especially in the era COVID and after COVID, where inactivity due to lack of physical fitness leads to a higher risk of falls
In the Mathematics and Methods section, the author describes in detail (phases) the adaptation of the FRQ tool. The feasibility and internal consistency checks were tested on 89 subjects and verified with the Cronbach's alpha score
Ethical approval was granted by the Ethics Committee of the Escola Superior de Saude da Cruz Vermelha Portugesa.
Semantic equivalence was clearly presented in Table 1.
The questions were revised by a panel of experts and some changes were made to questions that were unclear to the subjects.
The results are presented in a clear consultation format.
Reply: Thanks for your evaluation and comments.
Comment: The most important remark is that the discussion section and the comparison with similar studies and adaptations are missing.
Reply: Thanks for your suggestion. We improved the discussion part and believed that is better. Hope it meets your expectations.
Comment: Adapt the keywords according to MeSH.
Reply: Thanks for your suggestion. We preform the MeSH adaptation.
Comment: Is the manuscript clear, relevant for the field and presented in a well-structured manner? Please see the comments above.
Are the cited references mostly recent publications (within the last 5 years) and relevant? Does it include an excessive number of self-citations? Yes but it can improved becaue the discussion parti s missing.
Is the manuscript scientifically sound and is the experimental design appropriate to test the hypothesis? Yes
Are the manuscript’s results reproducible based on the details given in the methods section? Yes
Are the figures/tables/images/schemes appropriate? Do they properly show the data? Are they easy to interpret and understand? Is the data interpreted appropriately and consistently throughout the manuscript? Please include details regarding the statistical analysis or data acquired from specific databases. Yes
Are the conclusions consistent with the evidence and arguments presented? Yes
Please evaluate the ethics statements and data availability statements to ensure they are adequate.
Ethical approval has been obtained for testing the adapted version of the questionnaire.
Reply: Thanks for your evaluation and comments.
General questions to help guide your review report for review articles:
Is the review clear, comprehensive and of relevance to the field? Is a gap in knowledge identified? Yes,
there is a need to screen the population for falls as this is a major public health challenge.
Was a similar review published recently and, if yes, is this current review still relevant and of interest to the scientific community? no.
Are the statements and conclusions drawn coherent and supported by the listed citations? Yes.
Are the figures/tables/images/schemes appropriate? Do they properly show the data? Are they easy to interpret and understand? Yes.
During the manuscript evaluation, please rate the following aspects:
Novelty: Is the question original and well-defined? Do the results provide an advancement of the current knowledge? With suggested improvements it can provide advancement in the current research in the field of falls in older people..
Reply: Thanks for your assistance.
Scope: Does the work fit the journal scope*? Yes.
Significance: Are the results interpreted appropriately? Are they significant? Are all conclusions justified and supported by the results? Are hypotheses carefully identified as such? Yes.
Quality: Is the article written in an appropriate way? Are the data and analyses presented appropriately? Are the highest standards for presentation of the results used? Yes.
Interest to the Readers: Are the conclusions interesting for the readership of the journal? Will the paper attract a wide readership, or be of interest only to a limited number of people? (Please see the Aims and Scope of the journal.) Yes.
Overall Merit: Is there an overall benefit to publishing this work? Does the work advance the current knowledge? Yes with the improvements.
English Level: Is the English language appropriate and understandable? Yes
Reply: Thanks for your evaluation and suggestions. We hope the changes made meets your expectations.
Best regards.
Reviewer 4 Report
The conclusions should indicate practical solutions that are useful at work.
Author Response
Reply to Reviewer 4
Comment: The conclusions should indicate practical solutions that are useful at work
Reply: Thanks for your comment. Changes were made to the conclusions that we believe, meets your expectations.
Best regards
Reviewer 5 Report
Eligible for major revision, provided that (1) the urgency of the research needs to be strengthened; (2) the literature review is not clear; and (3) references must be added that are up-to-date and relevant to the research topic
Author Response
Reply to Reviewer 5
Comment: Eligible for major revision, provided that (1) the urgency of the research needs to be strengthened;
Reply: Thanks for your comment. We believe that the changes made improved the manuscript in this issue. Hope it meets your expectations.
Comment: (2) the literature review is not clear; and
Reply: Thanks for your comment. We believe that the changes made improved the manuscript in this issue. Hope it meets your expectations.
Comment: (3) references must be added that are up-to-date and relevant to the research topic
Reply: Thanks for your comment. We believe that the changes made improved the manuscript in this issue. Hope it meets your expectations.
Round 2
Reviewer 5 Report
Articles have been revised according to comments from reviewer.